# A Novel Approach to Investigate the Superheating Grain Refinement Process of Aluminum-Bearing Magnesium Alloys Using Rapid Solidification Process

**DOI:** 10.3390/ma16134799

**Published:** 2023-07-03

**Authors:** Sungsu Jung, Yongho Park, Youngcheol Lee

**Affiliations:** 1Energy Component & Material R&BD Group, Korea Institute of Industrial Technology, Busan 46938, Republic of Korea; jungss@kitech.re.kr; 2Department of Materials Science and Engineering, Pusan National University, Busan 46241, Republic of Korea

**Keywords:** AZ91 alloys, grain refinement, melt superheating, rapid solidification, Al-Mn particles

## Abstract

The superheating process is a unique grain refining method found only in aluminum-containing magnesium alloys. It is a relatively simple method of controlling the temperature of the melt without adding a nucleating agent or refining agent for grain refinement. Although previous studies have been conducted on this process, the precise mechanism underlying this phenomenon has yet to be elucidated. In this study, a new approach was used to investigate the grain refinement mechanism of aluminum-containing magnesium alloys by the melting superheating process. AZ91 alloy, a representative Mg-Al alloy, was used in the study, and a rapid solidification process was designed to enable precise temperature control. Temperature control was successfully conducted in a unique way by measuring the temperature of the ceramic tube during the rapid solidification process. The presence of Al_8_Mn_5_ and Al_10_Mn_3_ particles in non-superheated and superheated AZ91 ribbon samples, respectively, manufactured by the rapid solidification process, was revealed. The role of these Al-Mn particles as nucleants in non-superheated and superheated samples was examined by employing STEM equipment. The crystallographic coherence between Al_8_Mn_5_ particles and magnesium was very poor, while Al_10_Mn_3_ particles showed better coherence than Al_8_Mn_5_. We speculated that Al_10_Mn_3_ particles generated by the superheating process may act as nucleants for α-Mg grains; this was the main cause of the superheating grain refinement of the AZ91 alloy.

## 1. Introduction

Magnesium alloys are among the lightest metals, featuring low density and high specific strength properties [1]. These characteristics are considered as important factors in various industrial fields, particularly in aerospace, automotive, and electronics industries where weight reduction is crucial for specific applications [2]. The utilization of magnesium alloys for creating lightweight and robust structures can reduce fuel consumption, enhance energy efficiency, and decrease environmental impact [3]. However, the strength and ductility of magnesium alloys may be lower compared to other lightweight materials, such as aluminum alloys, which highlights the need to develop methods to enhance the strength and ductility of magnesium alloys in order to make them more competitive in various applications [3].

Grain refinement in magnesium alloys has proven to be an efficient method for enhancing strength and ductility simultaneously [4]. Research on the grain refinement of magnesium alloys is primarily divided into two categories: aluminum-free and aluminum-bearing magnesium alloys [5]. Zirconium is recognized as the most potent grain refiner for aluminum-free magnesium alloys. Qian et al. asserted that the zirconium-rich core structures in Mg-Zr alloys serve as nucleation sites for α-Mg grains, emphasizing zirconium’s remarkable grain refining capabilities [6]. However, zirconium forms stable compounds with Al, Mn, Si, Fe, Sn, Ni, Co, and Sb, so it cannot serve as a nucleating agent in magnesium alloys containing aluminum, which are mostly used in commercial applications [7]. Various techniques have been reported for the grain refinement of aluminum-containing magnesium alloys, but, unlike the addition of zirconium in aluminum-free magnesium alloys, there are no commercially available and reliable grain-refining processes for aluminum-containing magnesium alloys. Additionally, the mechanisms of grain refinement in these alloys have not been fully elucidated [8]. Therefore, further research is required to understand the grain refinement mechanisms and develop effective grain refining processes for aluminum-containing magnesium alloys.

One of the grain refinement mechanisms for aluminum-containing magnesium (Mg-Al) alloys is the superheating process, which enables grain refinement through a relatively simple method of controlling only the temperature of the melt without the addition of nucleating or refining agents [9]. In this process, the molten alloy is heated to a temperature well above its liquidus temperature (150–260 °C or higher) for a brief period, after which it is rapidly cooled down to the pouring temperature and held for a short time before being cast [10]. The effectiveness of superheating as a grain refinement method depends on the alloy’s composition, particularly its aluminum content. Alloys with higher aluminum content tend to exhibit better grain refinement when subjected to superheating. Other elements, such as Fe, Mn, and Si, can also influence the grain refinement effect, but excessive amounts of these elements can be counterproductive [11]. The key to successful grain refinement through superheating lies in determining the optimal temperature range and holding time. Once these optimal conditions are achieved, further increasing the holding time or repeating the process does not result in additional grain refinement. Rapid cooling and immediate pouring after reaching the superheating temperature are essential to prevent grain coarsening [12].

Although grain refinement by superheating has been known for a long time, the exact mechanism has not yet been identified. Some hypotheses have been proposed to explain the effect, but they all have their limitations. The oxide nucleation theory [13] suggests that grain refinement occurs through the formation of magnesium oxide and other oxide particles produced in the molten metal during the superheating process. However, grain refinement is observed even in a vacuum atmosphere where oxide formation is difficult. Moreover, it cannot explain why grain refinement occurs with compounds composed of Al, C, Fe, and Mn. Temperature-solubility theory [14] posits that particles too large for good nucleants at normal pouring temperatures are dissolved during superheating and re-precipitated as finer particles that act as nucleation sites. However, the theory does not identify specific particle species. Al_4_C_3_ particle nucleation theory [15] suggests that grain refinement in Mg-Al alloys occurs due to the nucleation of α-Mg grains on Al_4_C_3_ particles formed during the superheating process. Superheating causes the diffusion of carbon atoms from the steel crucible into the Mg molten metal, leading to the formation of Al_4_C_3_ particles, which are believed to be nucleation sites for α-Mg grains. Motegi’s study [16] on commercial AZ91E alloys showed the presence of numerous Al_4_C_3_ particles after superheating, supporting this hypothesis. However, there is no direct experimental evidence to confirm the formation of Al_4_C_3_ particles during the superheating process. Al-Mn intermetallic compound nucleation theory [17] proposes α-Mg grain nucleation by Al-Mn intermetallic compounds precipitated during cooling after superheating. It has been proposed that aluminum contributes to reducing the solubility of manganese in Mg melts, leading to the formation of various Al-Mn intermetallic compounds during the superheating process. Byun et al. [18] suggested that Al_8_Mn_5_ compounds provided heterogeneous nucleation sites for primary α-Mg grains. However, the edge-to-edge matching model by Zhang et al. [19] indicated that Al_8_Mn_5_ had low nucleating efficiency as nucleation sites for α-Mg grains. Cao et al. [20] reported ε-AlMn as apotent nucleant for α-Mg grains due to its presence in the Al-60Mn master alloy, which showed high grain refining efficiency. Qin et al. [21] supported these findings, showing that single-phase ε-AlMn refined AZ31 grains, while single-phase Al_8_Mn_5_ did not. In contrast, Qiu et al. [22] suggested that metastable τ-AlMn, which could be generated during the melt superheating process, had better crystallographic matching with the Mg matrix than ε-AlMn and acted as a nucleant for α-Mg grains. However, no direct evidence currently exists to confirm the presence of τ-AlMn and ε-AlMn in Mg-Al alloys, and methods to control their phase and morphology are still unknown. Duplex nucleation theory [9,10] suggests that Al_4_C_3_ particles are responsible for grain refinement in Mg-Al alloys, while Mn and Fe may interfere with the nucleation potency of Al_4_C_3_ particles. During the superheating process, Mn and Fe are dissolved, allowing Al_4_C_3_ particles to act as nucleants for α-Mg grains. However, prolonged holding at the pouring temperature causes Mn and Fe to re-wrap around the Al_4_C_3_ particles, coarsening the grains. Han et al. [23] supported these findings, showing that Al_4_C_3_ particle clusters act as nucleants for α-Mg grains rather than individual particles. In Mn-containing AZ91 alloy, the attachment of Al_8_Mn_5_ particles to Al_4_C_3_ reduces nucleation efficiency. Nevertheless, Al_4_C_3_ particles not attached to Al_8_Mn_5_ within a nucleating cluster can still play an effective role in refining a-Mg grains. Thus, it is deduced that Al_4_C_3_ particle clusters are beneficial in overcoming the hindering effect of Mn. However, definitive evidence regarding the formation of duplex structures involving Al_8_Mn_5_ and Al_4_C_3_ has not yet been established.

Previous studies have proposed various theories to explain the mechanism of grain refinement by superheating in Mg-Al alloys. However, these studies have certain shortcomings, such as a lack of direct experimental evidence to validate the theories and incomplete understanding due to the inability of any one theory to fully explain all observed phenomena. In this study, the rapid solidification process was employed to identify the nucleants generated by the superheating process. This approach was specifically designed to identify and analyze potential nucleants that may exist as solid phases within the molten Mg-Al alloy during the superheating process. By harnessing the rapid solidification process, these solid particles could be effectively captured and scrutinized, allowing for a deeper and clearer understanding of their structure and composition. Therefore, this study presents new perspectives on the mechanism of superheating grain refinement that have not been reported in any previous studies.

## 2. Materials and Experiments

### 2.1. Materials and Casting Process

AZ91D alloy, which is most widely used in industrial fields, was used in this study. Approximately 1000 g of the alloys were melted in an alumina crucible using an electric furnace. Samples were prepared with and without the superheating process. For samples without superheating, the AZ91D alloy was melted at 670 °C under a protective gas of 1.0% SF_6_ and 99.0% N_2_ and then poured into a mild steel mold preheated to 250 °C. For samples subjected to the superheating process, the alloy was melted at 670 °C under a protective gas, then rapidly heated to 770 °C and held at this temperature for 15 min, followed by rapid cooling to 670 °C, and finally poured into a mild steel mold preheated to 250 °C. The chemical composition of the samples was analyzed by an optical emission spectrometer (Spectro MAXx, SPECTRO, Kleve, Germany).

### 2.2. Rapid Solidification Process

Rapidly solidified ribbon samples were prepared using the melt spinner equipment (MSE 170, Yein Tech, Seoul, Republic of Korea) shown in Figure 1a. Rapidly solidified ribbon samples were prepared by melting 2 g of AZ91D alloy within a ceramic tube and subsequently injecting the molten metal into a rapidly rotating copper wheel at 1500 rpm. As shown in Figure 1b, each sample was melted using a ceramic crucible that has little reactivity with Mg molten metal. Cracking of the ceramic crucibles due to thermal shock was prevented by wrapping them with SUS304 tube. The thermocouple was attached to a ceramic crucible and was set at a position not affected by the induction heating coil. To enhance the contact between the ceramic crucible and the thermocouple, a copper sheet was used to wrap around the thermocouple. The high thermal conductivity of copper enabled the temperatures of the molten metal and crucible to reach an equilibrium state over time, ensuring proper melting. The process temperature was recorded by a data logger (NI cDAQ-9174, National Instruments, Austin, TX, USA) at the frequency of 20 Hz. Conventional casting samples were produced by melting the molten metal at a temperature of 650 °C, followed by a spraying process onto the rotating wheel. In contrast, superheating samples were prepared by first heating the molten metal to a temperature of 800 °C, then rapidly cooling it down to 650 °C before proceeding with the spraying process.

### 2.3. Measurement of Grain Size

Metallographic specimens were polished and then etched using an acetic acid-picral etchant to obtain a clear color contrast that enabled detailed investigation of the microstructure via optical microscopy. Microstructural analysis was performed using a high-resolution optical microscope (Leica MC 170, Leica, Teaneck, NJ, USA) to capture images of the specimen surfaces. The average grain size of the specimens was measured by focusing on the center of the cross-section of each specimen.

### 2.4. Thermal Analysis

Thermal analysis experiments were carried out using cylindrical graphite crucibles to determine the undercooling during the solidification of the alloys. The graphite crucible was submerged in the melt until its temperature reached the temperature of the molten metal. The crucible, filled with molten metal, was then transferred to a ceramic board. A K-type thermocouple, calibrated using the equilibrium melting temperature of high-purity (99.99%) aluminum, was inserted into the center of the melt to monitor the temperature throughout the solidification process. Cooling curves were recorded using a data logger (NI cDAQ-9174, National Instruments, Austin, TX, USA) at a frequency of 20 Hz.

### 2.5. Analysis of Microstructure on Rapidly Solidified Sibbon Samples

The microstructure of samples was analyzed using a field emission scanning transmission electron microscope (Talos F200X, Thermo Fisher Scientific, Waltham, MA, USA) after preparing the samples with focused ion beam (Scios2, Thermo Fisher Scientific, Waltham, MA, USA) equipment. The sample was first protected with a thin layer of Pt, then FIB milling was performed to create thin samples. Energy-dispersive x-ray spectroscopy (EDS, Thermo Fisher Scientific, Waltham, MA, USA) was employed to obtain more detailed elemental information, and high-resolution images were acquired to study the microstructure of samples. Crystallographic analysis of the phases observed in each sample was carried out utilizing Gatan software (ver 3.43) and CrysTbox software (ver 1.10).

## 3. Results

### 3.1. Microstructure and Chemical Composition

Figure 2 shows the effect of the superheating process on the microstructure and grain size of AZ91D alloy. The average grain size of the non-superheated sample was measured to be 310 µm, while the grain size decreased to 108 µm after the superheating process. Both samples exhibited dendritic microstructures with equiaxed grains, but the superheated sample had grains that were approximately three times smaller than the non-superheated sample. Additionally, the superheated sample showed enhanced uniformity in grain size and shape compared to the non-superheated sample, which had non-uniform grain shapes and sizes.

As shown in Table 1, the composition of Al, Zn, and Mn of both alloys is remarkably similar, thus indicating that the possibility of grain refinement effects resulting from constitutional undercooling can be excluded. Furthermore, because both specimens are manufactured under identical casting conditions, a more accurate comparison can be made, eliminating the potential for grain refinement effects due to differences in cooling rates. The consistency in casting conditions and composition ensures that the observed differences in microstructure can be ascribed to the superheating process, rather than variations in the manufacturing process.

### 3.2. Cooling Curve and Undercooling

Figure 3 illustrates the cooling curves of the non-superheated and superheated AZ91 alloys. The non-superheated AZ91 alloy exhibited an undercooling of 2.1 °C, whereas the superheated AZ91 alloy showed negligible undercooling. This suggests that the superheated AZ91 alloy contained effective nucleants that could initiate nucleation with minimal activation energy [24]. Cooling rates were measured at 1.1 °C and 0.9 °C for the non-superheated and superheated samples, respectively, and it was found that the impact of cooling rates on the observed undercooling variations was negligible. Therefore, it can be concluded that the observed differences in microstructure between the two samples can be attributed to the superheating process, rather than variations in cooling rates [25].

### 3.3. Design of the Precise Temperature Measurement System for the Rapid Solidification Process

In this study, the rapid solidification process was used to detect nucleants within the molten metal that play a significant role in grain refinement. However, measuring the temperature of the molten metal during rapid solidification using melt spinner equipment can be challenging due to the absence of a dedicated temperature measurement system and thermocouple inaccuracies caused by induction coil influence. To overcome this challenge, a method was designed to precisely measure the temperature of the molten metal, and this method was used to manufacture rapidly solidified AZ91 ribbon samples, as shown in Figure 1. By integrating this precise temperature measurement system into conventional melt spinner equipment, it is possible to produce rapidly solidified AZ91 ribbon samples with a precisely controlled superheat process. Figure 4a displays the temperature profile of the non-superheated and superheated AZ91 ribbon samples during the manufacturing process, while Figure 4b illustrates the resulting rapidly solidified AZ91 ribbon samples. The non-superheated AZ91 ribbon sample was heated at a rate of 0.43 °C/s until it reached the target temperature of 650 °C. The molten metal was then immediately sprayed onto the rapidly rotating copper wheel, which led to the formation of rapidly solidified samples. For the superheated AZ91 ribbon sample, a two-stage heating process was employed. First, the sample was heated to 454 °C at a rate of 0.41 °C/s. Then, the heating rate was increased and the sample was rapidly heated to 800 °C at a rate of 1 °C/s. Once the desired superheating temperature was achieved, the molten metal was held at 800 °C for 180 s to ensure uniform temperature distribution. The molten metal was then cooled at a controlled rate of 0.9 °C/s until it reached the target temperature of 650 °C, at which point it was sprayed onto the copper wheel to produce rapidly solidified samples.

### 3.4. Cooling Rate Calculation for Rapid Solidification Process

Determining the cooling rate of ribbons fabricated by rapid solidification is essential for the identification and analysis of solid-phase nucleants present in the molten AZ91 magnesium alloy. The cooling rate significantly influences the formation and growth of solid-phase nucleants within molten metals. By establishing and controlling the cooling rate, a systematic investigation of the formation of nucleants and their contribution to grain refinement can be conducted more effectively. Figure 5 shows the cooling rate of AZ91 ribbon samples at different copper wheel speeds. The cooling rate of the rapidly solidified AZ91 magnesium alloy ribbon was determined using an equation derived from a previous study [26]. The thickness of the ribbon samples was measured at different copper wheel speeds, and the data were used to calculate the cooling rate for each. For a specimen with a thickness of 50 μm, the cooling rate reached 2.69 × 10^6^ °C/s, while for a specimen with a thickness of 165 μm, the cooling rate was notably lower at 2.47 × 10^5^ °C/s. It was observed that the cooling rate demonstrated a tendency to increase as the thickness of the specimen decreased. For optimal results in rapid solidification, a higher cooling rate is preferable. However, when the copper wheel speed exceeded 2000 rpm, it became challenging to produce ribbon samples with consistent size and shape, making microstructure analysis difficult due to their small dimensions. Consequently, the rapid solidification process was conducted with a copper wheel speed of 1500 rpm, resulting in a ribbon sample with a thickness of 85 μm.

### 3.5. EDS Analysis of Rapidly Solidified AZ91D Ribbon Samples

Figure 6a shows an image of a non-superheated AZ91 alloy ribbon sample obtained using bright-field transmission electron microscopy. The microstructure of the sample contains dispersed particles within the magnesium matrix. One of these particles, indicated by a yellow arrow, was selected for further analysis using energy-dispersive X-ray spectroscopy mapping (Figure 6b). The analysis revealed that the particle did not contain magnesium but had significant amounts of aluminum (Figure 6c) and manganese (Figure 6d). The atomic ratios of aluminum and manganese in the particle were found to be 63.54 and 36.46, respectively, based on EDS point analysis (Figure 6e). This composition ratio is consistent with the Al_8_Mn_5_ intermetallic compound, which was confirmed through further analysis of other particles in the sample. The results demonstrate that the particles observed in the non-superheated AZ91 alloy ribbon sample were Al_8_Mn_5_ intermetallic compounds [27]. Upon conducting further analysis of the other particles using the same method, results consistent with the previous observation were obtained.

The image in Figure 7a shows a bright-field transmission electron microscopy image of a superheated AZ91 alloy ribbon sample, revealing the presence of particles dispersed throughout the magnesium matrix. One particle, indicated by a yellow arrow, was chosen for detailed analysis using energy-dispersive X-ray spectroscopy mapping. The analysis confirmed the absence of magnesium in the particle (Figure 7b) but the presence of aluminum (Figure 7c) and manganese (Figure 7d), consistent with the particles observed in the non-superheated sample. However, the atomic ratios of aluminum and manganese in the particle of the superheated sample (Figure 7e) were different from those in the non-superheated sample (Figure 6e). Specifically, the particle in the superheated sample exhibited an atomic ratio of 78.34 for aluminum and 21.66 for manganese, whereas the particle in the non-superheated sample had a ratio of 63.54 for aluminum and 36.46 for manganese. Based on these ratios, the particles were identified as Al_10_Mn_3_ in the superheated sample and Al_8_Mn_5_ intermetallic compounds in the non-superheated sample [27].

### 3.6. HR-TEM Images of Rapidly Solidified AZ91D Ribbon Samples

Table 2 shows the crystal structures of Mg, Al_8_Mn_5_, and Al_10_Mn_3_ [27]. It was observed that the crystal structure and lattice parameters of Al_8_Mn_5_ significantly diverge from those of Mg. Conversely, Al_10_Mn_3_ and Mg share the same crystal structure and even belong to the same space group; however, a distinction in their lattice parameters was observed. Considering the crystal structures of each phase, Al_10_Mn_3_ appears to present a higher propensity to function as a nucleat for α-Mg grains compared to Al_8_Mn_5_. However, to elucidate this with higher precision, it is essential to evaluate the interplanar spacings and investigate potential crystallographic mismatches between the planes interfacing with each phase.

Figure 8a shows a high-resolution transmission electron microscopy (HR-TEM) image of the magnesium matrix and Al_8_Mn_5_ particles in the non-superheated AZ91 alloy ribbon sample. A local inverse fast Fourier transform (IFFT) image of white circle A (Mg) is shown in Figure 8b. The spacing of white line in the IFFT image denotes the interplanar spacing of the region being measured. Using the IFFT image as a reference, a profile image was derived, as shown in Figure 8c, enabling the measurement of the white line distance. As there is a variance between the peak intervals, an average value was acquired by measuring over an extended distance, which resulted in a value of 0.2479 nm. Table 3 presents the planes and corresponding d-spacing (in nm) for Mg, Al_8_Mn_5_, and Al_10_Mn_3_, derived using the CrysTBox software (ver 1.10). The $\left(0\;1\;\bar{1}\;1\right)$ plane of Mg, with a D-spacing of 0.2453 nm, displayed the closest proximity to the 0.2479 nm value obtained through the IFFT profile image. The IFFT image of the white circle B, representing Al_8_Mn_5_, is shown in Figure 8d, and Figure 8e displays the profile image derived from this IFFT image. The Al_8_Mn_5_ particles were identified on the (1 4 1) plane and the interplanar spacing was calculated to be 0.2275 nm, using the same method employed for the determination of the plane and interplanar spacing of Mg. Through the analysis of the HR-TEM image, IFFT images were acquired for the contact area between magnesium and Al_8_Mn_5_ particles, leading to the identification of the magnesium plane as $\left(0\;1\;\bar{1}\;1\right)$ and the Al_8_Mn_5_ plane as (1 4 1). Furthermore, the interplanar spacing was determined for both planes: the magnesium $\left(0\;1\;\bar{1}\;1\right)$ plane had an interplanar spacing of 0.2453 nm, while the Al_8_Mn_5_ particle (1 4 1) plane had a slightly smaller interplanar spacing of 0.2275 nm.

Figure 9a shows a high-resolution transmission electron microscopy (HRTEM) image of the magnesium matrix and Al_10_Mn_3_ particles in the superheated AZ91 alloy ribbon sample. Local inverse fast Fourier transform (IFFT) images of two white circles, A and B, are also shown in Figure 9b,d, respectively. In addition, Figure 9c,d are profile images derived through IFFT images of two white circles, A and B, respectively. In the profile images of Mg and Al_10_Mn_3_, the average distance between the peaks was calculated to be 0.2406 nm and 0.2328 nm, respectively. The values obtained from the profile images were compared with the D-spacing values listed in Table 3, allowing for the confirmation of the Mg and Al_10_Mn_3_ planes. In the superheated sample, the magnesium plane was identified as $\left(0\;1\;\bar{1}\;1\right)$, which is consistent with the non-superheated sample. However, the Al_10_Mn_3_ plane was observed as (1 2 1), which is distinct from the Al_8_Mn_5_ plane found in the non-superheated sample. Furthermore, a detailed examination of the interplanar spacing was conducted, revealing an interplanar spacing of 0.2347 nm for the Al_10_Mn_3_ particle (1 2 1) plane in the superheated sample.

## 4. Discussions

The microstructure of as-cast AZ91D samples was investigated to determine the effect of superheating, and the results are presented in Figure 2. The non-superheated alloy exhibited an average grain size of 310 µm, while superheating reduced it to 108 µm. This indicates that the superheating process has a significant impact on the grain size of the alloy. Furthermore, the superheated AZ91D alloy had a more uniform and finer grain size than the non-superheated sample. The average grain size of the superheated sample was approximately three times smaller than that of the non-superheated sample. The chemical composition of the cast AZ91D samples was analyzed and is shown in Table 1. The primary additives, Al, Zn, and Mn, were found to have similar compositions in both alloys, thereby eliminating constitutional undercooling as a possible cause of grain refinement. In addition, because the samples were manufactured under the same casting conditions, any grain refinement effect due to the difference in cooling rate can be excluded. Thus, the observed difference in grain size between the non-superheated and superheated samples can be attributed to the superheating process. The cooling curves of non-superheated and superheated AZ91 alloys are shown in Figure 3. The extent of undercooling during solidification can reveal the presence of nucleants in the melt. When potent nucleants are present in the melts, the thermodynamic driving force required for the transition from the liquid phase to the solid phase is decreased, facilitating the formation of solid-phase nuclei with minimal undercooling [24]. This suggests that the cooling curves of the non-superheated and superheated AZ91 alloys provide information about the extent of undercooling during solidification, which can reveal the presence of nucleants in the melt. When potent nucleants are present, the thermodynamic driving force required for the transition from liquid to solid phase is decreased, facilitating the formation of solid-phase nuclei with minimal undercooling. Although a quantitative analysis of the nucleants was not conducted in this study, the degree of undercooling measured during solidification provides a basis for qualitatively deducing that the superheated AZ91 alloy possessed more nucleants promoting grain refinement than the non-superheated AZ91 alloy.

The rapid solidification process was used in this study to investigate the effect of nucleants on grain refinement in molten metal, as it can detect nucleants present in the molten metal. Ribbon samples were fabricated using non-superheated and superheated AZ91 alloys to study the impact of the superheating process on grain refinement and the contribution of nucleants to the formation of α-Mg grains. Rapid solidification also allows for the preservation of metastable phases within the molten metal, which is important for understanding the underlying mechanisms of grain refinement during the superheating process. Accurately measuring the temperature of the molten metal during the production of rapidly solidified ribbon samples using conventional melt spinner equipment is a significant challenge in this study. This is because precise temperature measurement is crucial to understanding the mechanism of grain refinement induced by the superheating process. However, conventional melt spinner equipment lacks an integrated temperature measurement system, and obtaining precise temperature control using a thermocouple is difficult due to the interference of the induction coil responsible for heating the metal material. In this study, a system capable of precise temperature measurement was designed and used to produce these rapidly solidified ribbon samples by modifying the existing melt spinner equipment. The temperature profiles during the fabrication of non-superheated and superheated AZ91 ribbon samples are presented in Figure 4a. It is evident that both samples were manufactured using a rapid solidification process at a precisely controlled temperature. The cooling rate is an important factor in identifying and analyzing solid-phase nucleants in AZ91 magnesium alloy produced by rapid solidification. It significantly affects the formation and growth of solid-phase nucleants, and controlling it allows for a more systematic investigation of their formation and contribution to grain refinement. The study by Wang et al. [26] investigated the heat transfer during rapid solidification of a ribbon prepared by the melt spinning process. They used a one-dimensional heat conduction equation to model the heat transfer, which allowed them to determine the temperature distribution and cooling rate within the ribbon. The cooling rate was found to be inversely proportional to the square of the thickness of the ribbon [26], and the same principle was applied for estimating the cooling rate of those rapidly solidified AZ91 alloy ribbon samples in this study. Various factors such as melting temperature, thermal conductivity, specific heat capacity, density, and latent heat were adjusted for the AZ91 magnesium alloy. The molten metal was rapidly solidified by spraying it onto a copper wheel rotating at a speed of 1500 rpm, resulting in the manufacturing of a ribbon sample with a thickness of 85 μm. This approach builds upon the findings of previous research and allowed for a systematic investigation of the cooling rate and its effect on the formation of solid-phase nucleants and subsequent grain refinement in the AZ91 magnesium alloy.

Bright-field transmission electron microscopy images of ribbon samples from non-superheated and superheated AZ91 alloys are shown in Figure 6a and Figure 7a, respectively. In both samples, particles are dispersed throughout the magnesium matrix. The EDS results showed that the particle in the non-superheated sample was an intermetallic compound with a composition of Al_8_Mn_5_, while the particle in the superheated sample was identified as Al_10_Mn_3_. Although manganese is an element added to enhance the corrosion resistance of the AZ91 alloy [28], it reacts with aluminum to form Al-Mn intermetallic compounds. Among these, Al_8_Mn_5_ is known to form prior to the primary magnesium phase [29], and its role as a nucleant for α-Mg grains has been debated in numerous studies. However, no definitive mechanism has been established [17,30]. One widely accepted theory is the edge-to-edge matching model proposed by Zhang et al. [19], which suggests that, crystallographically, Al_8_Mn_5_ cannot act as a nucleant for α-Mg grains. There are various theories regarding the mechanism of the superheating process, and one of them is related to the Al-Mn intermetallic compound nucleation [17]. Cao et al. [20] found that ε-AlMn, which is present in Al-60Mn master alloy, can act as a potent nucleant for α-Mg grains, showing excellent grain refinement efficiency. This was further supported by Qin et al. [21], who reported that single-phase ε-AlMn could refine AZ31 grains, whereas single-phase Al_8_Mn_5_ could not. However, Qiu et al. [22] suggested that metastable τ-AlMn, potentially formed during the melt superheating process, could exhibit better crystallographic matching with the Mg matrix than ε-AlMn, making it a more effective nucleant for α-Mg grains. Nonetheless, there is currently no clear evidence to verify the presence of both τ-AlMn and ε-AlMn in Mg-Al alloys, and there are no established methods to regulate their phase and morphology. The finding of a new particle, Al_10_Mn_3_, during the superheating process of the AZ91 alloy is significant as it has not been previously reported in the literature. This new particle may provide novel insights into the mechanism of grain refinement during the superheating process. However, further research is needed to fully understand the thermodynamic calculations and physical and chemical processes involved in the formation of Al_10_Mn_3_ during the superheating process.

The coherent relationship between two phases occurs when their interatomic distances and atomic arrangements on their crystal faces are similar, enabling one phase to act as a highly efficient heterogeneous nucleation site for the other phase [31]. It is important to note that for a favorable coherent relationship to exist at the interface of the two phases, comparable interplanar distances are essential. Classical nucleation theory suggests that a substrate’s ability to promote nucleation depends on the interfacial energy between the substrate and the nuclei. This interfacial energy is influenced by the crystallographic structures of both the substrate and the nuclei [32]. Turnbull and Vonnegut [33] proposed a one-dimensional misfit model to describe the relationship between substrate particle effectiveness and crystallographic mismatch for nuclei. However, this model has limitations when the crystallographic structures of the nuclei and substrate particles differ significantly. Bramfitt [34,35] improved upon Turnbull’s model by creating a two-dimensional model that takes into account the angle between the crystal orientations of both phases. This two-dimensional model enables the calculation of two-dimensional lattice misfit for matching planes.

In this modified Turnbull’s model, the two-dimensional lattice misfit of two matching planes was calculated based on Equation (1),
(1)$$\delta_{{\left(hkl\right)}_{n}}^{(hk{l)}_{s}}=\sum_{i=1}^{3}\frac{\left|\left(d_{{\left[uvw\right]}_{s}^{i}}cos\theta \right)-d_{{\left[uvw\right]}_{n}^{i}}\right|/d_{{\left[uvw\right]}_{n}^{i}}}{3}\times 100\%$$
where (h k l)_s_ is a plane of the substrate, [uvw]_s_ a direction in (h k l)_s_, (h k l)_n_ a plane of the nucleated solid, [uvw]_n_ a direction in (h k l)_n_, d[uvw]_s_ the interatomic distance along [uvw]_s_, d[uvw]_n_ the interatomic distance along [uvw]_n_, and θ is the angle between the [uvw]_s_ and [uvw]_n_.

This study found that there is crystallographic coherence between the interfaces of Al_8_Mn_5_ and Al_10_Mn_3_ particles with the magnesium matrix. The high-resolution transmission electron microscopy (HR-TEM) and inverse fast Fourier transform (IFFT) images of the non-superheated AZ91 alloy ribbon samples are shown in Figure 8, which revealed that the contact planes for magnesium and Al_8_Mn_5_ particles were the magnesium plane $\left(0\;1\;\bar{1}\;1\right)$ and the Al_8_Mn_5_ plane (1 4 1). Similarly, the HR-TEM and IFFT images of the superheated AZ91 alloy ribbon samples are shown in Figure 9, which identified the contact planes for magnesium and Al_10_Mn_3_ particles as the magnesium plane $\left(0\;1\;\bar{1}\;1\right)$ and the Al_10_Mn_3_ plane (1 2 1). The modified Turnbull’s model is effective in analyzing the coherence of two planes when their crystallographic structures are not significantly different. However, in the case of the (1 4 1) plane of Al_8_Mn_5_ and the $\left(0\;1\;\bar{1}\;1\right)$ plane of Mg, their crystallographic coherence was poor, and thus the application of the modified Turnbull’s model was not feasible. On the other hand, the (1 2 1) plane of Al_10_Mn_3_ and the $\left(0\;1\;\bar{1}\;1\right)$ plane of Mg showed good crystallographic coherence, and the modified Turnbull’s model could be used to analyze their coherence. Figure 10 shows the typical planar atomic arrangements in the (1 2 1) plane of Al_10_Mn_3_ and in the $\left(0\;1\;\bar{1}\;1\right)$ plane of Mg. The brown dash circles represent the Al_10_Mn_3_ atoms in the (1 2 1) plane and the blue circles represent the Mg atoms in the $\left(0\;1\;\bar{1}\;1\right)$ plane. The Al_10_Mn_3_ and Mg atoms applied to the lattice misfit analysis are indicated by filling the center with the same color as the circle, and the place where the Al_10_Mn_3_ and Mg atoms exactly match is marked in red. The *i*_1_ indicate a line connecting the arrangement of Al_10_Mn_3_ atoms in the [1 0 1] direction with an interatomic distance of 1.224 nm and a line connecting the arrangement of Mg atoms in the [3 3 −3] direction with an interatomic distance of 1.084 nm; the angle (θ) between both lines is 4°. The *i*_2_ and the *i*_3_ are marked in the same way as the *i*_1_; the required parameters for Equation (1) are listed in Table 4. The disregistry, δ, for Mg $\left(0\;1\;\bar{1}\;1\right)$║Al_10_Mn_3_ (1 2 1) is as follows:$$\delta_{{\left(1\;2\;1\right)}_{{Al}_{10}{Mn}_{3}}}^{\left(0\;1\;\bar{1}\;1\right)Mg}=\frac{\left(|1.081-1.244|/1.244)+(|2.956-3.21|/3.21)+(|1.734-1.524|/1.524\right)}{3}\times 100\%\;\approx 11\%$$

This indicates that the Al_10_Mn_3_ particle is a better nucleating site for α-Mg grains than the Al_8_Mn_5_ particle, and α-Mg can nucleate on the Al_10_Mn_3_ nucleation substrates. Previous research has shown that a substrate can effectively act as a nucleant for a solid if the disregistry between the two is less than 5% [36]. However, this is not always the case. Despite an 11% disregistry between Al_10_Mn_3_ particles and Mg, it is not conclusive that Al_10_Mn_3_ particles cannot act as nucleants for α-Mg grains. This study found that Al_8_Mn_5_ particles, which have low crystallographic coherence with Mg, can transform into Al_10_Mn_3_ through superheating, allowing them to act as nucleants for α-Mg grains. This finding provides a new perspective on investigating the mechanism of superheating grain refinement in the AZ91 alloy.

This study investigated the mechanism of superheated grain refinement in the AZ91 magnesium alloy through rapid solidification with precise temperature control. The results showed that Al_8_Mn_5_ may transform into Al_10_Mn_3_ during the superheating process. At a higher temperature, the solubility of Mn in the Mg melt will be increased and that would also affect the phase transformation of Mn intermetallics. From the HR-TEM and IFFT images of the samples, the Al_10_Mn_3_ can effectively act as a nucleant for grain refinement in AZ91. The formation of Al_10_Mn_3_ particles was found to play a crucial role in the superheating grain refinement process of the alloy. However, the study did not provide objective thermodynamic evidence to support the phase transformation of Al_8_Mn_5_ to Al_10_Mn_3_ or the generation of Al_10_Mn_3_, which will be explored in future research. Overall, this study offers a new perspective on the superheating grain refinement mechanism in AZ91 and provides insights for future research.

## 5. Conclusions

In this work, the mechanism on superheating grain refinement of the AZ91 alloy was investigated using a rapid solidification process. The main conclusions can be summarized as follows:(1)The average grain size of the non-superheated sample manufactured by the mold casting methods was measured to be 310 µm, while the grain size decreased to 108 µm after the superheating process.(2)The non-superheated AZ91 alloy exhibited an undercooling of 2.1 °C, whereas the superheated AZ91 alloy showed negligible undercooling. This suggests that nucleants, which influence the refinement of α-Mg grains, were generated by the superheating process.(3)Through the utilization of a rapid solidification process with precise temperature control, Al_8_Mn_5_ particles were observed in the non-superheated AZ91 ribbon samples; the (1 4 1) plane of these particles and the $\left(0\;1\;\bar{1}\;1\right)$ plane of magnesium was found to be in contact. However, it was confirmed that the crystallographic coherence between the two planes was so inconsistent that the modified Turnbull–Vonnegut equation, which is used for quantitative crystallographic coherence analysis, could not be applied.(4)Al_10_Mn_3_ particles were observed in the superheated AZ91 ribbon samples; the (1 2 1) plane of these particles and the $\left(0\;1\;\bar{1}\;1\right)$ plane of magnesium was found to be in contact. The 11% mismatch between the two planes was calculated using the modified Turnbull–Vonnegut equation.(5)It is thought that the superheating process contributes to grain refinement of AZ91 alloy by generating Al_10_Mn_3_, which exhibits more good crystallographic matching with magnesium compared to Al_8_Mn_5_. However, the study did not provide objective thermodynamic evidence to support the phase transformation of Al_8_Mn_5_ to Al_10_Mn_3_ or the generation of Al_10_Mn_3_, and additional thermodynamic studies are planned to clarify our results.

## Figures and Tables

**Figure 1 materials-16-04799-f001:**
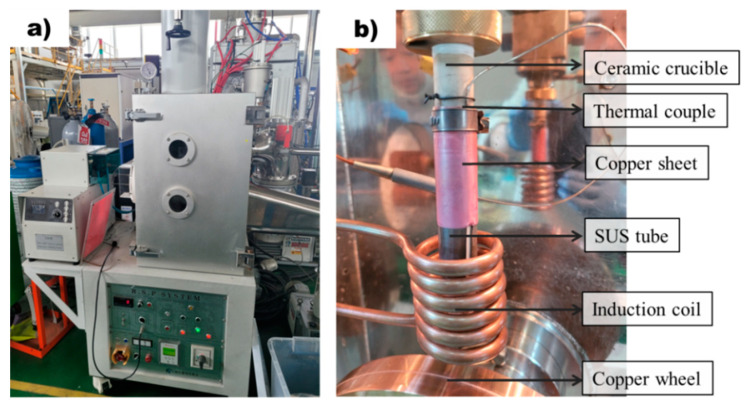
Apparatus for the rapid solidification process: (**a**) Equipment for rapid solidification process. (**b**) Rapid solidification process system with precise temperature control.

**Figure 2 materials-16-04799-f002:**
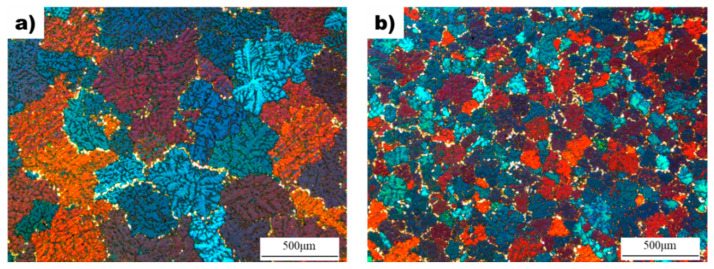
The microstructure of the as-cast samples: (**a**) Non-superheated AZ91D alloy. (**b**) Superheated AZ91D alloy with superheating.

**Figure 3 materials-16-04799-f003:**
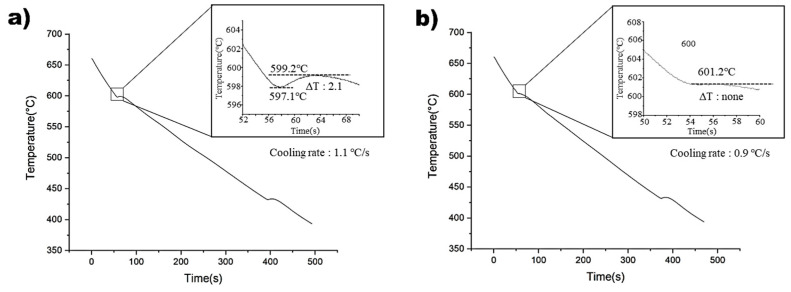
Cooling curves of: (**a**) Non-superheated AZ91 alloy. (**b**) Superheated AZ91 alloy.

**Figure 4 materials-16-04799-f004:**
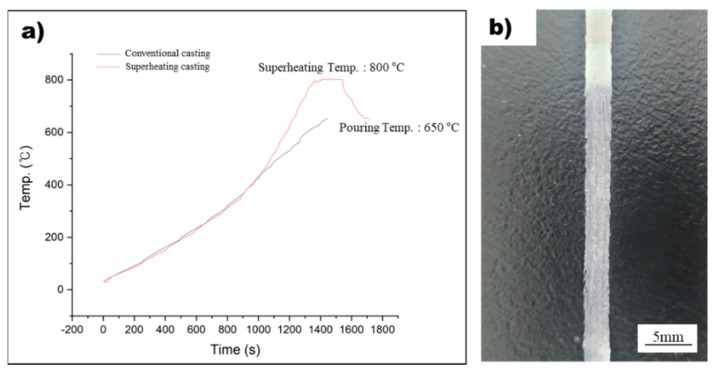
(**a**) The temperature profile during the manufacturing of non-superheated and superheated AZ91 ribbon samples. (**b**) The rapidly solidified AZ91 ribbon samples.

**Figure 5 materials-16-04799-f005:**
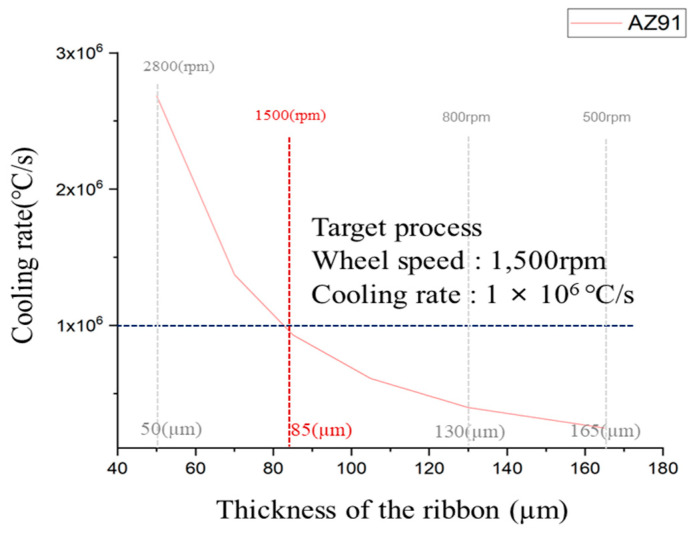
Cooling of AZ91 ribbon samples according Cu wheel to rpm.

**Figure 6 materials-16-04799-f006:**
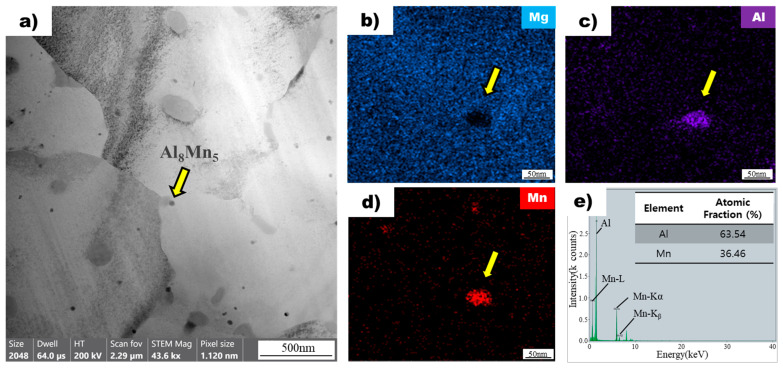
(**a**) Bright field TEM image of non-superheated AZ91 ribbon samples. The yellow arrow points to the Al_8_Mn_5_ particle. EDS mapping profile pointed by the yellow arrow of Mg (**b**), Al (**c**), and Mn (**d**) for the particle observed in non-superheated AZ91 ribbon samples. (**e**) EDS point analysis of the particle observed in non-superheated AZ91 ribbon samples.

**Figure 7 materials-16-04799-f007:**
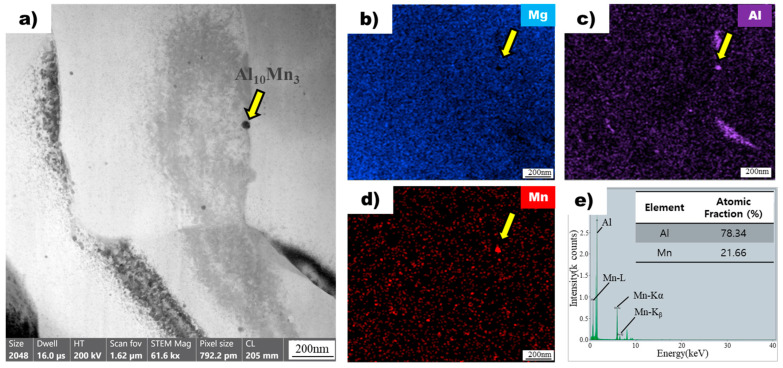
(**a**) Bright field TEM image of superheated AZ91 ribbon samples. The yellow arrow points to the Al_8_Mn_5_ particle. EDS mapping profile pointed by the yellow arrow of (**b**) Mg, (**c**) Al, and (**d**) Mn for particle observed in superheated AZ91 ribbon samples. (**e**) EDS point analysis of particle observed in superheated AZ91 ribbon samples.

**Figure 8 materials-16-04799-f008:**
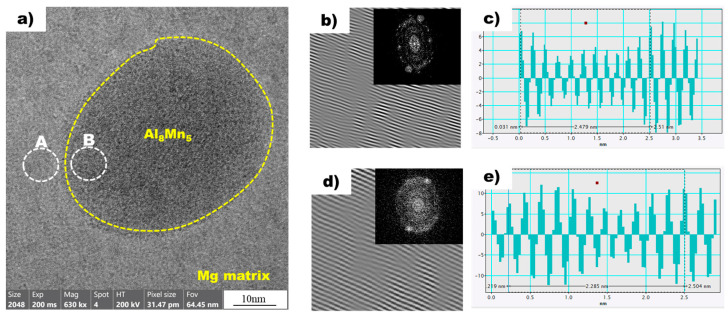
(**a**) HRTEM image of Mg (background) and Al_8_Mn_5_ (yellow dot circle) in the non-superheated AZ91 ribbon sample. (**b**) The local IFFT image of the white circle A (Mg) in (**a**). (**c**) The profiles of the local IFFT image (**b**). (**d**) The local IFFT image of the white circle B (Al_8_Mn_5_) in (**a**). (**e**) The profiles of the local IFFT image (**d**).

**Figure 9 materials-16-04799-f009:**
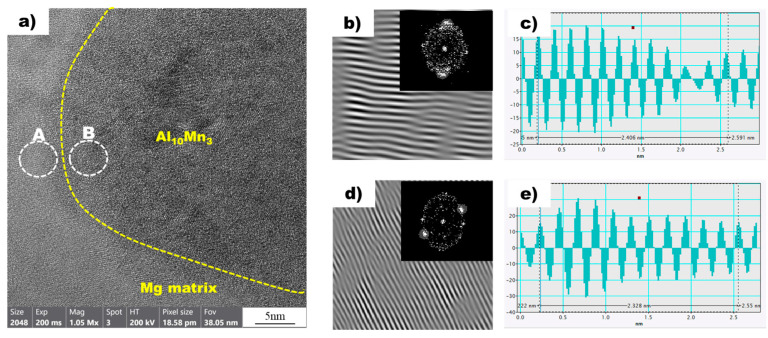
(**a**) HRTEM image of Mg (background) and Al_10_Mn_3_ (yellow dot circle) in the non-superheated AZ91 ribbon sample. (**b**) The local IFFT image of the white circle A (Mg) in (**a**). (**c**) The profiles of the local IFFT image (**b**). (**d**) The local IFFT image of the white circle B (Al_10_Mn_3_) in (**a**). (**e**) The profiles of the local IFFT image (**d**).

**Figure 10 materials-16-04799-f010:**
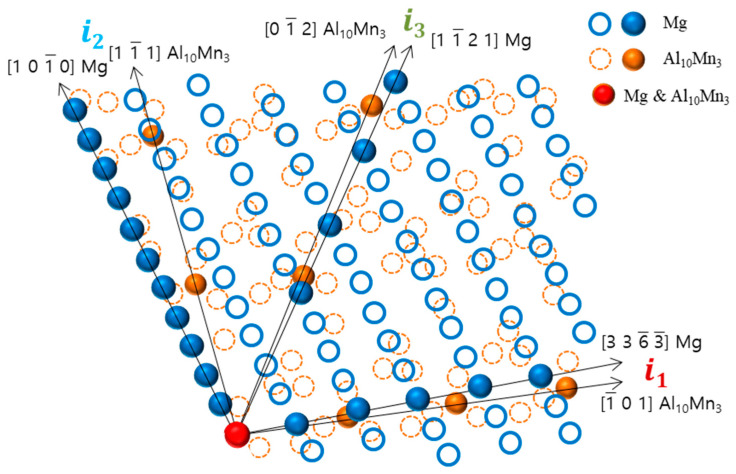
The crystallographic relationship at the interface between the (1 2 1) plane of Al_10_Mn_3_ and the $\left(0\;1\;\bar{1}\;1\right)$ plane of Mg.

**Table 1 materials-16-04799-t001:** Chemical compositions on non-superheated and superheated AZ91D alloy.

Alloy	Al	Zn	Mn	Si	Fe	Cu	Ni	Mg
Non-superheated AZ91D	8.93	0.57	0.250	0.015	0.0012	0.0016	0.0012	Bal.
Superheated AZ91D	8.88	0.58	0.251	0.015	0.0012	0.0018	0.0012	Bal.

**Table 2 materials-16-04799-t002:** The crystal structures for Mg, Al_8_Mn_5_, and Al_10_Mn_3_ [27].

Phase	Crystal Structure	Space Group	Space Group Number	Lattice Parameter (nm)	
Mg	Hexagonal	P6_3_/mmc	194	a and b = 0.320	c = 0.521
Al_8_Mn_5_	Rhombohedral	R3m	160	a and b = 1.264	c = 1.585
Al_10_Mn_3_	Hexagonal	P6_3_/mmc	194	a and b = 0.750	c = 0.783

**Table 3 materials-16-04799-t003:** List of Plane and D-Spacing (nm) for Mg, Al_8_Mn_5_, and Al_10_Mn_3_ (used CrysTbox ver1.10).

Magnesium		Al_8_Mn_5_		Al_10_Mn_3_	
Plane	D-Spacing (nm)	Plane	D-Spacing (nm)	Plane	D-Spacing (nm)
0 0 0 1	0.5210	0 0 1	0.7690	0 0 1	0.7789
¦	¦	¦	¦	¦	¦
0 0 0 2	0.2605	0 2 3	0.2320	0 1 3	0.2412
$0\;1\;\bar{1}\;1$	0.2453	1 4 1	0.2275	1 2 1	0.2347
$0\;1\;\bar{1}\;2$	0.1901	0 5 0	0.2183	0 3 0	0.2147
¦	¦	¦	¦	¦	¦
$5\;5\;\bar{10}\;3$	0.0316	5 5 3	0.1131	5 5 3	0.0772

**Table 4 materials-16-04799-t004:** Parameters for Equation (1).

Case	i	d[u v w]_s_	d[u v w]_n_	θ (°)
$\mathrm{Mg}\;\left(0\;1\;\bar{1}\;1\right)$ ║ Al_10_Mn_3_ (1 2 1)	1	1.084	1.224	4
	2	3.012	3.210	11
	3	1.736	1.524	2

## Data Availability

The data used to support the findings of this study are available from the corresponding author upon request.

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
