# Peer review of "A Novel Approach to Investigate the Superheating Grain Refinement Process of Aluminum-Bearing Magnesium Alloys Using Rapid Solidification Process"

_materials, 2023, doi:10.3390/ma16134799_

Round 1
Reviewer 1 Report
1. As authors showed, in the non-superheated and superheated AZ91 ribbon samples, Al8Mn5 and Al10Mn3 particles can be formed, but the results can not provied clear proof for this statement. In Figure 6 and figure 7, the EDS only shows the chemical composition of the particles but cannot identified as Al8Mn5 or Al10Mn3. please provide more results and analysis to confirm the formation of Al8Mn5 or Al10Mn3.
2. Figure 8 and Figure 9 are not clear and effective to show the formation of Al8Mn5 or Al10Mn3.
3. Both rapid solidification and Al8Mn5 or Al10Mn3 served as nucleants for the formation of α-Mg grains can lead to the grain refiement of AZ91, but the author thought that the formation of Al10Mn3 particles acted as effective nucleants for the α-Mg grains. But the grain refinement is not abvious and may not the main effect. Please provide more evidentiary results.
4. Formula is incorrectly formatted, e.g. formula 1 on page 12.
5. Data formats are inconsistent, e.g. crystal surface indices on page 13.
6. Calculation is incorrect, e.g. the triaxial crystal plane index (0 1 1)Mg on page 13 should be converted to a tetraaxial plane index of (0 1
1)Mg.
7. Using the edge-to-edge matching model one should firstly calculate the row spacing mismatch (fr) and the crystal plane mismatch (fd). Furthermore, the amount of data is too small.
8. The format of the writing is not standard, e.g. there is no space between Al10Mn3 particles in the fourth line of the second paragraph on page 13.
9. The abstract section is too long, the research purpose and methods are not clear, and the conclusion section lacks detailed data support.
10. There is too much literature review and there is no comprehensiveness.
11. The experimental section and 2.4 should be placed at the top, followed by various testing and analysis.
12. In the analysis and discussion section, if the theory proposed in the introduction section is not analyzed based on experimental results, there should be no literature review or formula layout issues.
Moderate editing of English language required.
Reviewer 2 Report
In this work, authors reported the superheating grain refinement of Al-beating Mg alloys by rapid solidification process. Authors found that the formation of Al10Mn3 particles was formed in the process of superheating. They proposed Al10Mn3 might act as the effective nucleants for the growth of α-Mg grains. It may be published in Materials after a revision.
Suggestion,
(1) In Fig. 5, is the curve of cooling rate vs. thickness the theoretical curve? How about the data from your experiments? Just one point of 85 µm? If so, the points from experiments are not enough.
(2) In Fig. 2, is the microstructure from SEM image? Authors should give more discussion about dendritic microstructures? Why is the grain size smaller in superheated sample?
(3) In Fig. 6 and 7, the scale bar in (b, c, d) is unclear. In e, the plot of EDS is not formal.
(4) For Fig. 8 and Fig. 9, authors should explain the data of TEM in detail. For example, how about the lattice constants in A and B region? How about crystal quality?
· There’s a lot of repetition between the part of discussions and the part of introduction. The language of this manuscript is not refined enough.
Reviewer 3 Report
This is good work and interesting work. Several minor comments are follows.
1. “....the clear mechanism of grain refinement in aluminum-containing magnesium alloys by melt super-heating process was investigated. “ Was it defined?
2. The atomic content for fine precipitates is shown with very large precision. Please, verify.
3. The figures required corrections. Make sizes of numbers uniform and larger. Scale bars and their captures are also unclear.
4. Check punctuation.
